# Curcumin Treatment Identifies Therapeutic Targets within Biomarkers of Liver Colonization by Highly Invasive Mesothelioma Cells—Potential Links with Sarcomas

**DOI:** 10.3390/cancers12113384

**Published:** 2020-11-16

**Authors:** Daniel L. Pouliquen, Alice Boissard, Cécile Henry, Stéphanie Blandin, Pascal Richomme, Olivier Coqueret, Catherine Guette

**Affiliations:** 1Université d’Angers, Inserm, CRCINA, F-44000 Nantes, France; olivier.coqueret@univ-angers.fr; 2Université d’Angers, ICO Cancer Center, Inserm, CRCINA, F-44000 Nantes, France; alice.boissard@ico.unicancer.fr (A.B.); cecile.henry@ico.unicancer.fr (C.H.); catherine.guette@ico.unicancer.fr (C.G.); 3Université de Nantes, Plate-forme MicroPICell, SFR François Bonamy, F-44000 Nantes, France; stephanie.blandin@univ-nantes.fr; 4Université d’Angers, SONAS, EA921, SFR QUASAV, F-49000 Angers, France; pascal.richomme@univ-angers.fr

**Keywords:** metastatic tumor cells, sarcomatoid phenotype, malignant mesothelioma, proteomics, biomarkers, invasiveness, liver colonization, curcumin

## Abstract

**Simple Summary:**

Aggressive sarcomatoid tumors designed in inbred strains of immunocompetent rats represent useful tools for both the identification of biomarkers of invasiveness and evaluation of innovative therapies. Our aim was to investigate the molecular determinants of liver colonization and potential common biomarkers of sarcomas and sarcomatoid tumors, using the most invasive (M5-T1) of our four experimental models of peritoneal sarcomatoid malignant mesothelioma in the F344 rat. Using an advanced and robust technique of quantitative proteomics and a bank of paraffin-embedded tumor and tissue samples, we analyzed changes in the proteotype patterns of the liver from normal rats, adjacent non-tumorous liver from untreated tumor-bearing rats, and liver from tumor-bearing rats positively responding to repeated administrations of curcumin given intraperitoneally. The identification of proteome alterations accounting for the antitumor effects of curcumin and changes in the liver microenvironment, which favored the induction of an immune response, could be useful to the research community.

**Abstract:**

Investigations of liver metastatic colonization suggest that the microenvironment is preordained to be intrinsically hospitable to the invasive cancer cells. To identify molecular determinants of that organotropism and potential therapeutic targets, we conducted proteomic analyses of the liver in an aggressive model of sarcomatoid peritoneal mesothelioma (M5-T1). The quantitative changes between SWATH-MS (sequential window acquisition of all theoretical fragmentation spectra) proteotype patterns of the liver from normal rats (G1), adjacent non-tumorous liver from untreated tumor-bearing rats (G2), and liver from curcumin-treated rats without hepatic metastases (G3) were compared. The results identified 12 biomarkers of raised immune response against M5-T1 cells in G3 and 179 liver biomarker changes in (G2 vs. G1) and (G3 vs. G2) but not in (G3 vs. G1). Cross-comparing these 179 candidates with proteins showing abundance changes related to increasing invasiveness in four different rat mesothelioma tumor models identified seven biomarkers specific to the M5-T1 tumor. Finally, analysis of correlations between these seven biomarkers, purine nucleoside phosphorylase being the main biomarker of immune response, and the 179 previously identified proteins revealed a network orchestrating liver colonization and treatment efficacy. These results highlight the links between potential targets, raising interesting prospects for optimizing therapies against highly invasive cancer cells exhibiting a sarcomatoid phenotype and sarcoma cells.

## 1. Introduction

Recent studies on metastatic colonization have confirmed the hypothesis that certain tissue microenvironments may be preordained to be intrinsically hospitable to disseminated cancer cells [1]. The liver is the organ most frequently involved in this process, and the bidirectional interactions between cancer cells and the unique hepatic niche have revealed that both parenchymal and nonparenchymal cells modulate each step of the invasion–metastasis cascade [2]. In particular, the role of tumor–host interactions has been investigated [3], emphasizing the importance of microenvironments promoting factors in the initial formation and growth of micrometastases in this organ [4]. The need to better understand the molecular determinants of that organotropism has recently been reiterated [5]. This research field is also crucial in the context of cancer treatment to identify key molecular players which could also represent potential therapeutic targets.

A mechanism known as epithelial–mesenchymal transition (EMT) helps cancer cells acquire an increased migratory and invasive capacity, which partially explains the liver metastasis of some carcinomas [4]. These cancers are generally characterized by a great diversity of ‘molecular portraits’ [6]. In the case of breast cancer, study of the different intrinsic subtypes has benefited from the recent development of robust quantitative proteomics techniques [7]. To pinpoint which molecular alterations determine the occurrence of the most lethal, metaplastic subtype of breast cancer, Djomehri et al. have recently used this approach to identify the proteomic landscape that characterizes the sarcomatoid phenotype [8]. Cases of ‘carcinosarcoma’, ‘sarcomatous transformation of the connective tissue stroma of carcinoma’, and ‘spindle-cell metaplasia of epithelial cells’ were discussed by Greenblatt many decades ago [9]. Sarcoma-like tumors, characterized by their aggressiveness, have been documented in the breast, thyroid, uterus, and kidney among others, but are also represented by one of three different histological subtypes of malignant mesothelioma (MM), showing the worst median survival [10]. Moreover, intrapleural and intraperitoneal MM and sarcomas have both been established in laboratory animals following induction with polycyclic aromatic hydrocarbons [11] or refractory ceramic fibers [12]. Interestingly, common biomarkers between sarcomatoid MM [13] and sarcomas [14] have just started to be identified.

To investigate both the molecular determinants of liver colonization and potential common biomarkers of sarcomas and sarcomatoid MM, we used the most invasive of our four experimental rat sarcomatoid MM models [15]. This model previously led us to identify S100A4 as a proteomic biomarker of all stages of MM pathogenesis [16]. It was obtained after i.p. (intraperitoneal) injection of a cell line (M5-T1) to syngeneic rats, established from an immunocompetent male F344 rat after 378 days of induction with crocidolite fibers injected intraperitoneally, the tumor being characterized by immunosuppression [15] and the rapid development of metastases in multiple organs and tissues lining the peritoneal cavity [17]. Moreover, the treatment of M5-T1 tumor-bearing rats with repeated intraperitoneal administrations of curcumin stopped the invasion process while producing an immune response directed against residual tumor cells [17]. This background allowed us to ascertain the molecular determinants of both liver invasion and curcumin therapeutic efficacy. Cross-comparing proteomic analyses of histological sections of the liver from normal rats, adjacent non-tumorous liver from untreated tumor-bearing rats, and liver from curcumin-treated rats without hepatic metastases identified a list of 179 biomarkers whose abundances specifically changed in untreated rats. Among these biomarkers, multiple correlations were observed with seven biomarkers of M5-T1 tumor aggressiveness and one biomarker of immune response, several of them having previously been reported in the literature on proteomic studies of sarcomas, raising interesting questions to be discussed.

## 2. Results

### 2.1. Histological Characterization of Lymphocyte Changes in the Liver of Curcumin-Treated Rats

Three groups of rats were used for the study, normal rats (“Control”) as a reference, rats with untreated M5-T1 tumor (“Tumor”), and rats with M5-T1 tumor positively responding to repeated administrations of curcumin given intraperitoneally (“Tumor + Curcumin”). The timescale and procedure used for the selection of liver tissue samples from the two groups “Tumor” and “Tumor + Curcumin” are illustrated in Figure 1A, and representative examples of histological features of rats selected for constituting the “Tumor” group, or excluded from it, are illustrated in Figure 1B–D.

We previously demonstrated that rats given four injections of 1.5 mg/kg of curcumin on days 7, 9, 11, and 14 after tumor challenge presented a significant reduction in their mean total tumor mass compared with untreated rats (Figure 1A) [17]. As a subgroup of four animals (Figure 1A, red ellipse in the graph, and Figure 1I–K)) among the group of curcumin-treated rats presented no metastases in the liver but increased numbers of circulating CD8 (cluster of differentiation 8)+ T cells, we first aimed to characterize the location, morphology, numbers, and sizes of lymphocytes present in their liver parenchyma in comparison with untreated rats and controls. Histological analysis of high-magnification views revealed a very significant increase in lymphocyte counts in the liver of curcumin-treated rats (Figure 1E) and a significant increase in their mean length compared both with normal rats and untreated rats with tumors (Figure 1F). Additionally, in contrast to lymphocytes from normal rats (Figure 1G) and untreated rats with tumors (Figure 1H) characterized by isolated, round-shaped lymphocytes, often vicinal to vessels, most lymphocytes of curcumin-treated rats were found moving into sinusoids converging from the liver capsule, some of them being in contact with residual isolated tumor cells (Figure 1I).

### 2.2. Biomarkers Involved in the Immune Response Directed against M5-T1 Cells

We detected 1411 proteins in each liver sample analyzed by SWATH-MS (sequential window acquisition of all theoretical fragmentation spectra). Comparison of the proteins for which differences in log2-fold changes of the mean were statistically significant (*p*-values < 0.05) between the three groups of rats revealed the number was 439 for G2 (untreated tumor-bearing rats) vs. G1 (normal rats), 549 for G2 vs. G3 (curcumin-treated rats without hepatic metastases), and 355 for G3 vs. G1. Cross-comparing the three files identified two common proteins, purine nucleoside phosphorylase (PNPH, encoded by *Pnp*) and ceruloplasmin (CERU, encoded by Cp). PNPH represents an enzyme involved in lymphocyte differentiation, the decreased content of which is associated with immunodeficiency. CERU is a ferroxidase enzyme, the major copper-carrying protein in the blood, and an acute-phase protein commonly elevated in malignant diseases. Among the 12 biomarkers of interest found in the three files, PNPH was the only one evolving differently in G3 and G2 relative to G1 (Figure 2A). CERU presented exactly the inversed pattern of changes compared with PNPH (Figure 2B). An increase specifically observed in G3 affected the immunity-related GTPase family M protein (IRGM, encoded by *Irgm*), which positively regulates autophagosome maturation and the IFN-γ (interferon-gamma)-mediated signaling pathway, the macrophage migration inhibitory factor (MIF, encoded by *Mif*), and the antigen peptide transporter 2 (TAP2, encoded by *Tap2*) (Figure 2C). This pattern of changes was also observed for the AA alpha chain of the RT1 (name of the Major Histocompatibility Complex of the rat) class I (HA12, encoded by *RT1-A^a^*), the intercellular adhesion molecule 1 (ICAM1, encoded by *Icam1*), and the vesicle-trafficking protein SEC22b (SC22B, encoded by *Sec22b*), all involved in the presentation of foreign antigens to the immune system and/or T-cell activation (Figure 2C). A different pattern, consisting of a specific decrease in G3, was shared by two proteins, the apolipoprotein A-IV (APOA4, encoded by *Apoa4*) and the mitochondrial 60-kDa heat shock protein (CH60, encoded by *Hspd1*), involved in multiple functions, including host immune response in line with lipid metabolism (Figure 2D). Two other proteins, decreased under curcumin treatment but showing a tendency to increase in untreated rats (relative to control), were the alpha-2-HS-glycoprotein (FETUA, encoded by *Ahsg*) and interleukin-6-receptor subunit beta (IL6RB, encoded by *Il6st*), involved in the regulation of inflammatory response and interleukin-6 binding (Figure 2E). In contrast, proteasome activator complex subunit 2 (PSME2, encoded by *Psme2*) and the GTP-binding protein SAR1b (SAR1B, encoded by *Sar1b*) remained at comparable levels in G3 vs. G1 while downregulated in G2, both being implicated in efficient antigen processing (Figure 2F). 

### 2.3. Biomarkers Specifically Related to Metastatic Colonization of the Liver by M5-T1 Cells

For this purpose, the two files of proteins exhibiting abundance changes (*n* = 439 for G2 vs. G1 and *n* = 549 for G2 vs. G3) were cross-compared, and then all proteins appearing significantly changed in the third list (*n* = 355 for G3 vs. G1) were excluded (Figure 3A). This led to a final list of 179 proteins, 61 presenting a common decrease and 118 showing a common increase in G2 vs. G1 and G2 vs. G3 while unchanged in G3 vs. G1. The proteins with the largest decreases are illustrated in Figure 3B (for NCPR (NADPH-cytochrome P450 reductase) located in the endoplasmic reticulum), Figure 3C (for CALM (Calmodulin) located in the cytoplasm and cytoskeleton), and 3D (for AATM (Aspartate aminotransferase, mitochondrial), ADT2 (ADP/ATP translocase 2), DECR 2, 4-dienoyl-CoA reductase, mitochondrial), and PRDX3 (Thioredoxin-dependent peroxide reductase, mitochondrial)located in the mitochondria). The proteins with the largest increases are illustrated in Figure 3E (for ACADL (Long-chain specific acyl-CoA dehydrogenase, mitochondrial), ASSY (Argininosuccinate synthase), AT1A1 (Sodium/potassium-transporting ATPase subunit alpha-1), and LASP1 (LIM and SH3 domain protein 1) specifically located in mitochondria, cytoplasm, or plasma membrane, respectively), Figure 3F (for MLEC (Malectin), PGRC1 (Membrane-associated progesterone receptor component 1), and RPN1 (Dolichyl-diphosphooligosaccharide-protein glycosyltransferase subunit 1) mainly located in the endoplasmic reticulum), and Figure 3G (for EF1A1 Elongation factor 1-alpha 1), EZRI (Ezrin), HSP7C (Heat shock cognate 71 kDa protein), and UFL1 (E3 UFM1-protein ligase 1) located in multiple subcellular compartments). The subcellular localization of all other proteins with abundance decreases or increases are illustrated in Figure 4A,B, respectively, their full names and genes encoding the corresponding proteins in *homo sapiens* being given in Appendix A. 

### 2.4. S100 Differential Abundance Proteins

As we demonstrated that S100A4 represented a crucial biomarker of tumor invasiveness and host organ colonization by M5-T1 cells [16], we next investigated the changes observed within S100 protein abundances in liver samples from G1, G2, and G3. Five different S100 proteins were detected, S10A4 (encoded by *S100a4*), S10A6 encoded by *S100a6*), S10A8 (encoded by *S100a8*), S10AA (encoded by *S100a10*), and S10AB (encoded by *S100a11*). For three proteins, a return to normal levels (no significant differences relative to the liver from normal rats) was observed under curcumin treatment, the abundance being increased for S10A6 and S10AB and decreased for S10A8 (Figure 5). In contrast, the pattern of change was different for S10A4 while no differences were observed for S10AA (Figure 5).

### 2.5. Common Biomarkers of Liver Colonization by M5-T1 Cells and the M5-T1 Tumor

To identify biomarkers common to liver colonization by tumor cells and the M5-T1 tumor, the list of 439 proteins identified in G2 vs. G1 was cross-compared with three lists of proteins showing significant log2-fold changes in protein content in (1) all invasive tumors [1 = F4-T2, 2 = F5-T1, and 3 = M5-T1] vs. the noninvasive [4 = M5-T2], (2) the two more invasive tumors [2 = F5-T1 and 3 = M5-T1] vs. the noninvasive [4 = M5-T2], and (3) the most invasive tumor [3 = M5-T1] vs. the noninvasive [4 = M5-T2]. The procedure used to establish these lists was described in a previous study [15]. The proteins shared by each condition are illustrated in Figure 6A. In total, seven proteins were specifically found to be common to the M5-T1 tumor: coronin-1A (COR1A, encoded by *Coro1a*), heat shock cognate 70-kDa protein 1b (HS71B, encoded by *Hspa1b*), proliferation-associated protein 2G4 (PA2G4, encoded by *Pa2g4*), subunit 1 of dolichyl-diphosphooligosaccharide-protein glycosyltransferase (RPN1, encoded by *Rpn1*), homolog 3 of mothers against decapentaplegic (SMAD3, encoded by *Smad3*), anionic trypsin-1 (TRY1, encoded by *Prss1*), and cationic trypsin-3 (TRY3, encoded by *Try3*). Two additional proteins involved in abundance changes and found to be common to the two most invasive tumors, M5-T1 and F5-T1, were also considered, exocyst complex component 4 (EXOC4, encoded by *Exoc4*) and calponin-3 (CNN3, encoded by *Cnn3*). Two additional MarkerView statistical analyses between M5-T1 vs. F5-T1 (3 vs. 2) and M5-T1 vs. F4-T2 (3 vs. 1) finally reduced the number to seven (Figure 6C), as proteins showing no significant differences (*p* > 0.05) in either situation were excluded from the list (Figure 6B). In addition to the biomarkers specifically shared by the M5-T1 tumor, a large number of proteins were also found to be common to liver colonization by M5-T1 cells and to the invasive nature of the three tumors (Figure 6A).

### 2.6. Correlations between Identified Biomarkers

We next analyzed the correlations between protein abundance changes within the G2 group (untreated rats) for the main biomarkers identified in Section 2.2 (PNPH), Section 2.3 and Section 2.4 (all 179 listed biomarkers), and Section 2.5 (COR1A, CNN3, HS71B, RPN1, SMAD3, TRY1, and TRY3). Figure 7A summarizes the most important correlations and Appendix A illustrates them (for *p*-values < 0.01, Spearman’s statistical test). Additional correlations for bigger *p*-values, 0.01 < *p* < 0.02 and 0.02 < *p* < 0.05, are illustrated in Appendix A, respectively.

As important differences were observed for some biomarkers in correlation plots illustrating changes in abundances within the eight rats composing the G2 group (“Tumor”), we next wondered if they could be related to the stage of tumor development defined in Figure 1B,C (advanced vs. initial stage), represented each by four rats. Analysis of the seven biomarkers common to the liver colonization by M5-T1 tumor cells and the most invasive tumor (see Section 2.5 and Figure 6) first revealed that only TRY1 and TRY3 exhibited significant differences between the two stages (Figure 7B). Secondly, the analysis of all other biomarkers identified in Section 2.2, Section 2.3 and Section 2.4 and illustrated in Figure 2, Figure 3, Figure 4 and Figure 5 showed that, as for trypsinogens, abundance decrease and increase were enhanced with tumor progression for three (ACADS, CHSP1, SARDH) and 16 additional proteins (ADHX, AL1L1, CO5A1, CP2B1, DNJC8, DX39A, FETUA, FIBG, GRPE1, IRGM, MMGT1, PDLI1, PELP1, PGRC1, RTCB, S10A6), respectively. Figure 7B illustrates the proteins concerned with the most significant differences (*p* values < 0.01) and Appendix A describes all other cases (for 0.01 < *p* < 0.05). For 17 other proteins (AP2S1, ARL2, DJB11, ECHA, FABP5, FINC, H2AJ, HMCS2, HNRPK, IDH3A, LYPA1, NCPR, PA2G4, PICAL, PPAC, SRSF2, VIME), change in abundance did not followed tumor progression (Appendix A).

Finally, as abundance variations within the G2 group (“Tumor”) could influence the previously described effect of curcumin on these two lists of 21 and 17 proteins, we calculated *p* values for the four comparisons “Initial Stage (G2) vs. Control (G1)”, “Advanced Stage (G2) vs. Control (G1)”, “Tumor + Curcumin (G3) vs. Initial Stage (G2)”, and “Tumor + Curcumin (G3) vs. Advanced Stage (G2)”. The results, which are summarized in Table 1 and Table 2, confirm that, among specific biomarkers of liver invasion by M5-T1 tumor cells, the most interesting is TRY1. Interestingly, among the 26 proteins which correlate with TRY1 (Appendix A), CHSP1, RTCB (Table 1), and ECHA, FABP5, LYPA1, and NCPR (Table 2) represent other unspecific candidate biomarkers of liver colonization by highly invasive tumor cells, which also appear to be affected by curcumin treatment.

## 3. Discussion

Since 2003, the number of publications on sarcomatoid tumor subtypes has risen continuously. Given the subtypes’ commonalities, the growing interest is probably explained by the parallel exponential increase in reports on the involvement of EMT in aggressive cancers and on the therapeutic challenges. In this study, we investigated the molecular determinants of liver colonization by M5-T1 sarcomatoid MM cells, which also represented potential therapeutic targets. The biomarkers of invasiveness raised several questions of interest relevant to the immunodeficiency induced by tumor cells through connections with the ER (endoplasmic reticulum)–Golgi, the dysregulation of cytoskeletal, extracellular matrixes, and heat shock proteins. The relationships with parallel findings on these biomarkers’ role in the invasiveness of different sarcoma subtypes were also discussed.

Among the proteins detected by SWATH-MS and potentially involved in the immune response induced by curcumin treatment, purine nucleoside phosphorylase was the only one exhibiting an opposite change in abundance in treated and untreated rats relative to normal rats. The decrease specifically observed in untreated rats suggests immunosuppression caused by tumor development, comparable to the immunodeficiency reported in early research on acquired immunodeficiency syndrome (AIDS) [18]. The characteristics of this enzyme have been extensively reviewed by Bzowska et al., emphasizing, in particular, the impact of its deficiency on impairment of T-cell function [19]. Moreover, this common immune deficiency observed in PNPH-deficient patients and AIDS has also been observed in PNPH knockout mice [20]. T-cell deficiency that results from altered purine metabolism pathways has led to the development of powerful PNPH inhibitors for therapies [19,21]. Pioneering findings on the role of PNPH in different types of sarcoma cell lines have also contributed to the study of purine analogs having both anticancer and immunosuppressive activities [22]. Interestingly, this research has pointed to the role of the gene coding for methylthioadenosine phosphorylase, often codeleted with *p16* (*CDKN2A*), particularly in MM and several types of sarcomas [23]. Our findings on the increased level of CERU in untreated rats and the beneficial effect of curcumin are also consistent with previous reports on the increased expression of this biomarker associated with the tumor progression of lung cancer and its prognosis, particularly the work of Matsuoka et al. [24]. Finally, regarding other molecules of interest that we identified, another interesting point was the positive effect of curcumin treatment on the decrease in HA12 and TAP2 that we previously observed in relation to the immunodeficiency produced by M5-T1 tumor cells [16].

Among the seven biomarkers specific to the M5-T1 tumor, coronin-1A (COR1A) was the only one exhibiting a decreased abundance in untreated rats. This protein, which is involved in the dynamics of the actin cytoskeleton in response to T-cell receptor stimulation and cell activation, was found to be highly expressed in solid lymphoma [25]. Mouse models have revealed that CORO1A downregulation results in the induction of T-cell anergy, while lymphocyte proliferative responses to mitogen stimulation were impaired in all coronin-1A-deficient patients [26]. In our study, the common feature of CORO1A and PNPH downregulation, which specifically characterized the group of untreated rats and was reversed in rats treated with curcumin, is consistent with our histological observations and tends to confirm that M5-T1 cells induce an immunodeficient environment favorable to invasion of the liver parenchyma.

Another cytoskeletal protein, calponin 3 (CNN3), was specifically overexpressed in untreated rats. This actin-binding protein associates with stress fibers, which contribute to the migration and mechanosensing of non-muscle cells [27]. Calponin expression has been documented in several reports focusing on different types of sarcomas [28,29], leading to the conclusion that CNN3 is required for controlling proper contractility of the stress fiber network [27]. Two recent reports have confirmed the interest of this protein, revealing its key role in both invasiveness and drug resistance in gastric [30] and colon cancer cells [31]. Finally, among the correlations found between the seven M5-T1 tumor biomarkers and the 179 biomarkers of liver colonization, two interesting observations were the links between the increase in calponin 3 and prohibitin (PHB) and between the increase in nucleoporin (NUP53) and decrease in coronin 1A. These results are also consistent with our previous finding that PHB and NUP53 exhibited an increase in abundance related to the acquisition of invasive properties, which was common to the three different models of rat MM [15].

We previously demonstrated that S10A4, a member of the S100 family of calcium-binding and proinflammatory proteins [32], represented a crucial biomarker of tumor invasiveness and host organ colonization by M5-T1 cells [15], a protein of equal interest for studies on osteosarcomas [33,34]. In this study, the combined increase in S10A6 and S10AB in untreated rats and return to normal values following curcumin treatment are consistent with previous findings that, among eight proteins belonging to this family, these two showed the most dramatic increase in tumor vs. normal tissues [35]. The differential pattern of changes observed in S10A4 relative to S10A6 could be explained by the immune response specifically observed in curcumin-treated rats as S10A4, in contrast to S10A6, is highly expressed in immune cells [36]. The absence of further difference in abundance of S10A6 in treated vs. normal rats also tends to confirm the impact of curcumin treatment on cancer cell motility given the involvement of S10A6 on cytoskeletal organization [37]. The return to normal values of S10AB under treatment is also consistent with our previous finding of a dramatic increase in this protein’s abundance specifically observed in neoplastic vs. preneoplastic rat mesothelial cells [15]. The comparable S10AB values observed in curcumin-treated and normal rats lead to the conclusion that the activation of different signaling pathways, which were associated with S10AB increases in different types of cancer [38,39,40,41], was cancelled under treatment. Finally, we found the same opposite evolution of S10A4 and S10A8 as that observed in oral squamous cell carcinomas, suggesting that combinatorial analysis of these two S100 proteins would predict the lesions’ degree of malignancy [42].

In parallel to S100 proteins, when overexpressed trypsinogens have been shown to represent predictors of distant metastasis and survival for some cancers [43]. Guilardi et al. have demonstrated that trypsin-3 (also named pretrypsinogen III), encoded by *Try3* in the rat and *PRSS3* in humans, is overexpressed under the combined action of VEGF-A (Vascular endothelial growth factor A) and FGF-2 (Fibroblast growth factor 2) in endothelial cells, thereby boosting their migration and tumor angiogenesis [44]. We previously revealed the upregulation of these two growth factors, which characterized the most invasive of our four rat MM models, M5-T1 [15]. Interestingly, herein we showed that one of the specific biomarkers of liver colonization by these tumor cells was TRY3, whose increased abundance in untreated rats is reversed in curcumin-treated rats. This observation corroborates the findings of Wang et al., showing that *PRSS3* overexpression is correlated with metastasis and poor prognosis in patients with gastric cancer [45]. Moreover, we observed a good correlation between the increased abundances of both this protein and S100A6, and an even better correlation with another trypsinogen, TRY1, encoded by *Prss1* in the rat and *PRSS1* in humans. Another interesting finding was the difference observed in TRY1 abundance in the advanced versus initial stage of M5-T1 tumor development, and the correlations observed with CHSP1, ECHA, FABP5, LYPA1, NCPR, and RTCB, among other proteins, which suggest they could represent important therapeutic targets for limiting the deleterious effect of invasive tumor cells on the liver. In particular, CHSP1 (calcium-regulated heat-stable protein 1), the abundance decrease of which was enhanced according to the tumor development stage, represents a negative regulator of liver gluconeogenesis [46], which could play an important role in the context of colonization of this organ by invasive tumor cells. The involvement of alterations of the *Prss1* gene has previously been associated with elevated gastric cancer risk [47] and pediatric tumors including sarcomas [48]. The production of tumor-associated trypsinogens by fibrosarcoma, among other tumors, was also earlier reported [49]. Another intriguing feature was the concomitant suppression of immunodeficiency (attested by the evolution of immune response biomarkers, in line with our previous findings on HA12 and TAP2 [15]) and the return to normal values of both trypsinogens in curcumin-treated rats, which appears to be connected to the question raised by Chen et al. on immune escape in pancreatic cancer [50]. A high correlation was also observed for both trypsinogens with the 2B1 isoform of cytochrome P-450, which could be explained by the liver inflammation [51] associated with the start of liver parenchyma invasion by M5-T1 cells. Several nuclear proteins also appeared to be associated through correlations with the increased abundance of the two trypsinogens, particularly HMGN5, whose involvement in nucleosomal binding and transcriptional activation in cancers has been well documented [52]. Finally, of interest was the observation of multiple correlations with the abundance changes of mitochondrial proteins, particularly the trifunctional enzyme hydroxyacyl-CoA dehydrogenase alpha subunit (encoded by the *HADHA* gene in humans). The inverse correlation observed with the protein encoded by *Prss1* was consistent with the recent demonstration of the key role of its downregulation, together with other mitochondrial lipid metabolic enzymes, in tumor growth [53].

Analysis of all the correlations observed between the seven biomarkers of interest and the 179 liver biomarkers changed in G2 vs. G1 and G3 vs. G2, but not in G3 vs. G1, and led to one protein correlated with four of them, the subunit gamma of the translocon-associated protein (SSRG, encoded by *Ssr3*). This protein, localized in the endoplasmic reticulum, is a member of the translocon-associated protein (TRAP) complex, which is required for protein translocation across the ER membrane following translation and is involved in immune signaling [54]. Another member of the TRAP complex has been reported to be associated with the acquisition of multidrug resistance in human osteosarcoma cells [55]. The dysregulation of the *SSR3* gene has also been found to be included in the transcriptomic profile of resting NK (Natural Killer) cells [56]. In addition to being correlated with abundance changes of TRY1 and PNPH, the increase in SSRG was associated with the increase in RPN1 (dolichyl-diphosphooligosaccharide-protein glycosyltransferase) and HS71B (heat shock cognate 70-kDa protein 1B, encoded by *HSPA1B* gene in human). RPN1 is involved in the N-glycosylation of proteins, whose dysregulation in cancers began to be deciphered at the molecular level during the last decade [57], with interesting prospects for innovative therapies [58]. Elevated levels of HS71B (also named Hsp 70-2) in cancer cells may be responsible for tumorigenesis and for tumor progression by providing resistance to chemotherapy. So this protein has been identified as a potential target in the development of cancer therapeutics [59]. With S100A4 and HMGBs (High Mobility Group Box proteins), it also represents a “damage-associated molecular pattern” (DAMP), released from injured or necrotic liver cells bound to Toll-like receptors to modulate inflammatory reactions [60]. Upregulation of the *HSPA1B* gene is linked to stress-response pathways [61], and its multiple interactions explain why correlations are also found with mitochondrial (AATM), cytoplasmic (GYS2, PUR6), ECM (B2MG), and nuclear proteins (DHRS4). The deregulation of glycogen metabolism in hepatocellular carcinoma was recently documented through the downregulation of GYS2 expression, which was correlated with unfavorable patient outcomes [62].

Finally, our study tends to confirm the potential of curcumin in the treatment of various cancers, which was previously emphasized by numerous reports, reviews, and books. For devastating cancers, like malignant mesothelioma, progress in the use of this phytochemical has also been reviewed by Baldi and colleagues [63], opening up interesting prospects for clinical translation. Moreover, although the half-life of curcumin is short, its effect on cancer cells is quick [17], and the benefits provided by local (intrapleural) vs. intravenous administration have just started to be explored, representing alternatives for the treatment of pleural-based tumors [64].

## 4. Materials and Methods

### 4.1. NMR Characterization of the Batch of Curcumin Used for Treatment

The source of curcumin (batch 046K0691, 99% purity) used in the group of curcumin-treated rats, as previously described in [17], was provided by Sigma-Aldrich (L’Isle-d’Abeau, Chesnes, France). Quality was controlled using NMR spectroscopy after diluting 5 mg of dry powder (vial preserved at −20 °C) in 600 µL of DMSO-d6. NMR analyses were performed at 298 K on a JEOL 400-MHz YH spectrometer (JEOL Europe, Croissy-sur-Seine, France) equipped with an inverse 5-mm probe (ROYAL RO5). For ^13^C NMR (100 MHz) spectra, a WALTZ-16 decoupling sequence was used with an acquisition time of 1.04 s (32,768 datapoints) and a relaxation delay of 2 s. Then, 12,000 scans were collected to obtain a satisfactory S/N (signal-to-noise) ratio. The 2D HMBC (Heteronuclear Multiple Bond Correlation, ^1^H observed heteronuclear long-range *J*_HC_ correlations) was run using the standard JEOL pulse sequence (hmbc.jxp, 32 scans). NMR spectra presented all typical ^1^H and ^13^C chemical shifts, as reported by Benassi et al. [65]. The 2D Heteronuclear Multiple-Bond Correlation (HMBC) experiment, which provided a chemical shift correlation map between ^1^H and ^13^C, revealed that the ketonic and enolic carbons were undistinguishable and equivalent, with a ^13^C *δ* value of 183.6 ppm (see Appendix A).

### 4.2. Collection of Tissues for Histological and Proteomic Analyses

The collection of tissue samples used for this study came from three series of experiments conducted on Fischer F344 rats, two normal rats (“Control”), eight rats bearing M5-T1 tumor (“Tumor”), and four rats with M5-T1 tumor treated by curcumin (“Tumor + Curcumin”). For the “Tumor” group, paraffin-embedded sections were selected among a large tumor bank established from untreated rats bearing M5-T1 tumor and euthanized from day 16 to day 21 for different past experiments. In a first step, attention was given to samples that contained liver tissue of significant size (more than 5 mm^2^) without important foci of metastatic tumor cells. In a second step, eight samples collected from rats exhibiting different stages of tumor development (diaphragm, peritoneum, and/or spleen invasion or not, mean size, and numbers of tumor nodules attached or infiltrating the liver, the pancreas, and gut [16,17]) were selected. Within this group, the “Initial Stage” and “Advanced Stage” of tumor development were represented by an equivalent number of four samples/rat. For the “Tumor + Curcumin” group, paraffin-embedded sections of samples that contained liver tissue of enough size and collected from four of the six rats treated with curcumin that positively responded to the treatment were used. These rats were also selected on the basis that they presented only small residual tumor masses (<600 mg) in the peritoneal cavity at necropsy (see Figure 9A, red arrow in [17]), together with evidences of infiltration of residual tumors by CD8+ T lymphocytes (see also Figure 1J,K).

### 4.3. Experimental Procedures for In Vivo Manipulations

The rats were purchased from Charles River Laboratories (L’Arbresle, France) and maintained under SPF (Specific-pathogen-free) status and standard conditions in the UTE-IRS UN (Unité de Thérapeutique Expérimentale de l’Institut de Recherche en Santé de l’Université de Nantes) animal facility in compliance with European Union guidelines for the care and use of laboratory animals in research protocols (experiments approved by the ethics committee for animal experiments (CEEA) of the Pays de la Loire Region, France, 2011.38 from 2011 to 2015 and approval #01257.03 of the French Ministry of Higher Education and Research (MESR) from 2015 to 2018). The neoplastic cell line M5-T1 was injected intraperitoneally (3 × 10^6^ cells in 200 µL of PBS (Phosphate-buffered saline)) at day 0 in syngeneic rats (“Tumor” and “Tumor + Curcumin” groups). For curcumin-treated tumor-bearing rats (six rats), four successive injections of 1.5 mg/kg curcumin (which corresponded to 150 µL of the 10 mM stock solution in DMSO, diluted in 0.3 mL sterile NaCl 0.9%) were given intraperitoneally on days 7, 9, 11, and 14 [17] and tissues were collected on day 16 and fixed in 4% paraformaldehyde in PBS. The rats were anesthetized using an isoflurane chamber (Forene^®^, Abbott, Rungis cedex, France) and euthanized with a rate of 30% volume displacement per minute of CO_2_ in their home cage.

### 4.4. Histology

For histological examination, the paraformaldehyde fixed, paraffin-embedded sections of rat liver samples from the three different groups were cut with a Bond-Max automaton (Menarini, Rungis, France) and stained with hematoxylin-phloxine-saffron (HPS). For the characterization of infiltrated lymphocytes, slides, which included liver sections and had been stained with anti-CD3 (SM253P, Acris antibodies, San Diego, CA, USA) and anti-CD8 (LS-B3665, LSBio France, Nanterre, France) monoclonal antibodies as previously described [17], were selected. Lymphocyte counts and dimensions were analyzed from scans of the slides (NanoZoomer 2.0 HT Hamamatsu, Japan) and examination at high magnification (×800). A total of 10 to 15 different views from at least three different samples was used to analyze each group of rats.

### 4.5. Sample Preparation for SWATH-MS Analysis

Paraffin-embedded 5-µm-thick sections of each liver sample, stained with HPS, were first examined to select areas of interest in the liver at distance from the tumors. Then, all the corresponding areas were removed with a scalpel from five thicker (20-µm) sections of the samples and collected in a microtube. The samples were deparaffinized, dried, and treated as previously described in [16], and salts were removed accordingly. Peptide concentrations of the samples were finally calculated using the Micro BCA^TM^ Protein Assay Kit (Thermo Fisher Scientific, Saint-Herblain, France).

### 4.6. Relative Quantification by SWATH Acquisition and Statistical Analysis

Each sample (5 µg) was analyzed as previously described [15,16]. The method consisted of repeating the whole gradient cycle, which corresponded to acquiring 32 TOF (Time-Of-Flight) MS/MS scans of overlapping sequential precursor isolation windows (25 m/z isolation width, 1 *m/z* overlap, high sensitivity mode) covering the 400 to 1200 *m/z* mass range, with a previous MS scan for each cycle. The accumulation time was 50 ms for the MS scan (from 400 to 1200 *m/z*) and 100 ms for the product ion scan (230 to 1500 *m/z*), giving a total cycle time of 3.5 s.

Peak extraction of the SWATH data was performed using either the Spectronaut software (v 8.0, Biognosys, Schlieren, Switzerland) or SWATH MicroApp embedded in PeakView (v 2.0, AB Sciex Pte, Ltd., Framingham, MA, USA). SWATH data were processed with default settings in Spectronaut. Reference peptides from the iRT (Indexed retention Time) Kit (Biognosys) spiked into each sample were used to calibrate the retention time of extracted peptide peaks using Spectronaut. Peptide identification results were filtered with a q-value of <1%, excluding shared peptides. RT calibration was also performed based on iRT peptide elution profiles in PeakView using the SWATH App module (v 2.0). After peak extraction with either Spectronaut or PeakView, the sum of MS2 ion peak areas of SWATH-quantified peptides for individual proteins were exported to calculate the protein peak areas.

For statistical analysis of the SWATH dataset, the peak extraction output data matrix from PeakView was imported into MarkerView (v 2, AB Sciex Pte, Ltd., Framingham, MA, USA) for data normalization and relative protein quantification. Proteins with a fold change >1.5 and statistical *p*-value < 0.05 estimated by MarkerView were considered to be differentially expressed under different conditions.

## 5. Conclusions

Our data suggest that curcumin produces proteome alterations accounting for its antitumor effects. The modifications observed in the liver microenvironment under treatment, which favored the induction of an immune response, involved a specific increase in the level of purine nucleoside phosphorylase, decrease in ceruloplasmin, and return to normal values of coronin 1A. Additional changes suggest that molecules participating in autophagosome maturation, regulation of inflammatory response, efficient antigen processing, and presentation to the immune system for T-cell activation are also important parts of this process. The parallel increased levels of calponin-3, some S100 proteins, trypsinogens, and heat shock protein cognate 70-kDa protein 1b specifically observed in untreated rats and their correlations with quantitative modifications of some key mitochondrial, nuclear, or ER–Golgi proteins represent interesting questions for future investigations. Finally, this study highlights existing links between potential targets that could involve both metastasizing cancer cells with a sarcomatoid phenotype and sarcoma cells.

## Figures and Tables

**Figure 1 cancers-12-03384-f001:**
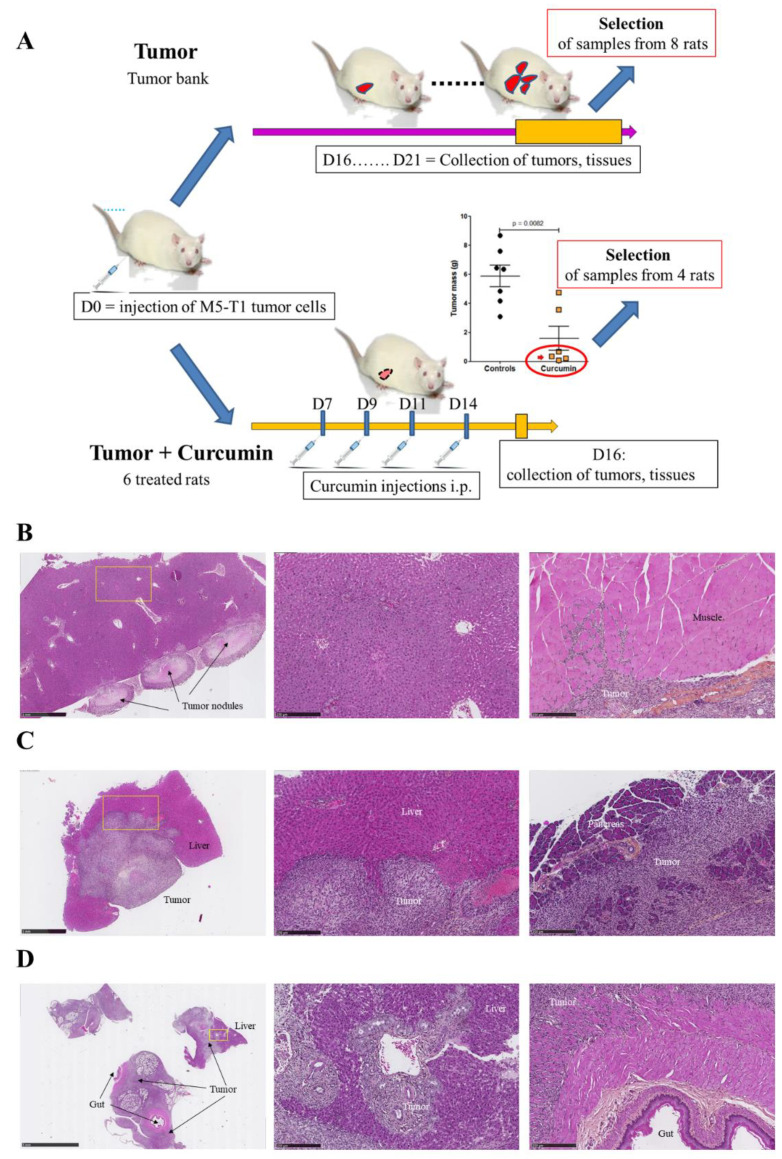
Experimental design and characteristics of the activated lymphocytes infiltrating the liver parenchyma of curcumin-treated rats. (**A**) Scheme of experimental design representing how liver samples from the “Tumor” and “Tumor + Curcumin” groups were selected for proteomic and histological analysis. (**B**,**C**) Histological features of two selected rats belonging to the “Tumor” group showing different stages of M5-T1 tumor development, initial stage (**B**), and advanced stage (**C**). Left, general views (×25, the scale bar represents 1 mm). Middle, magnification (×100, the scale bar represents 250 µm) of the yellow rectangles showing the histological features of the liver parenchyma used for proteomic analysis (at distance from the tumor front). Right, beginning of muscle invasion at initial stage (**B**) and invasion of the pancreas at advanced stage (**C**). (**D**) Example of rat at final stage from the tumor bank and excluded from sample selection. Left, general view (×6.5, the scale bar represents 5 mm), showing dramatic invasion of the liver and gut. Middle, magnification (×100, the scale bar represents 250 µm) of the yellow rectangle showing extended invasion of the liver parenchyma by tumor cells. Right, magnification (×100, the scale bar represents 250 µm) showing tumor cells invading the muscularis externa of the gut. (**E**) Quantification of lymphocytes in high-magnification fields. (**F**) Comparison of lymphocyte sizes. High-magnification views (×800, the scale bars represent 25 µm) of liver parenchyma from normal rats (**G**), rats with M5-T1 tumor ((**H**), tumor cells are indicated by yellow arrows), and rats with M5-T1 tumor treated with curcumin i.p. (**I**). Isolated tumor cells (green arrows, left and middle photographs in (**I**)) exhibiting morphological changes in comparison with (**H**), with numerous activated lymphocytes present in the sinusoids and converging toward the residual tumor cells. Two representative examples of immunohistochemical staining of CD3+ lymphocytes (**J**) and CD8+ lymphocytes (**K**) infiltrating the liver parenchyma of rats with M5-T1 tumor treated with curcumin i.p. (two different rats).

**Figure 2 cancers-12-03384-f002:**
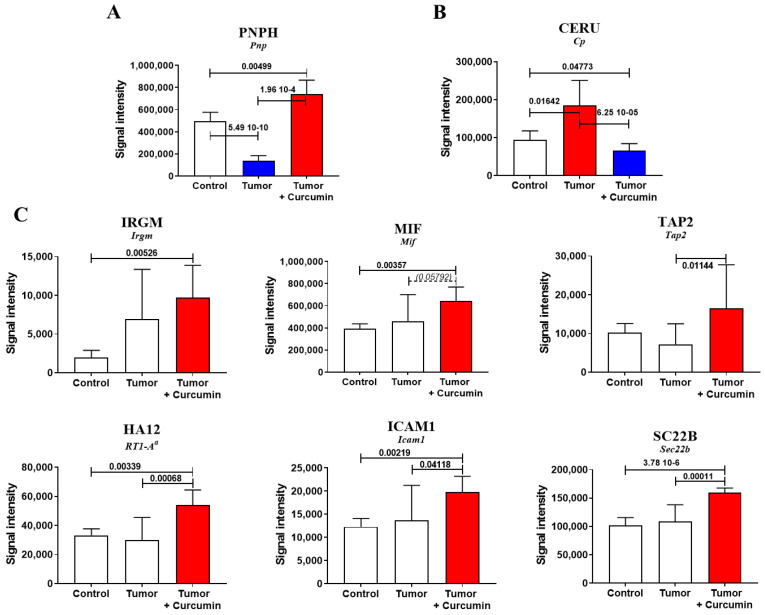
Biomarkers associated with the curcumin-induced immune response. Comparison of protein abundances between the three groups of rats, revealing different patterns of changes (significant increases in red and decreases in blue, with *p*-values indicated above the bars). Blank bars correspond to the absence of significant differences between groups. The names of genes coding for the proteins are indicated in italics below each protein abbreviation. (**A**), (**B**) differential evolution of protein abundances in treated (G3) and untreated rats (G2) relative to normal rats (G1). (**C**) Increased abundance in G3 (relative to G1 and/or G2). (**D**) Decreased abundance in G3 (relative to G1 and G2). (**E**) Decreased abundance in G3 (relative to G2). (**F**) Decrease in G2 vs. G1 and G2 vs. G3.

**Figure 3 cancers-12-03384-f003:**
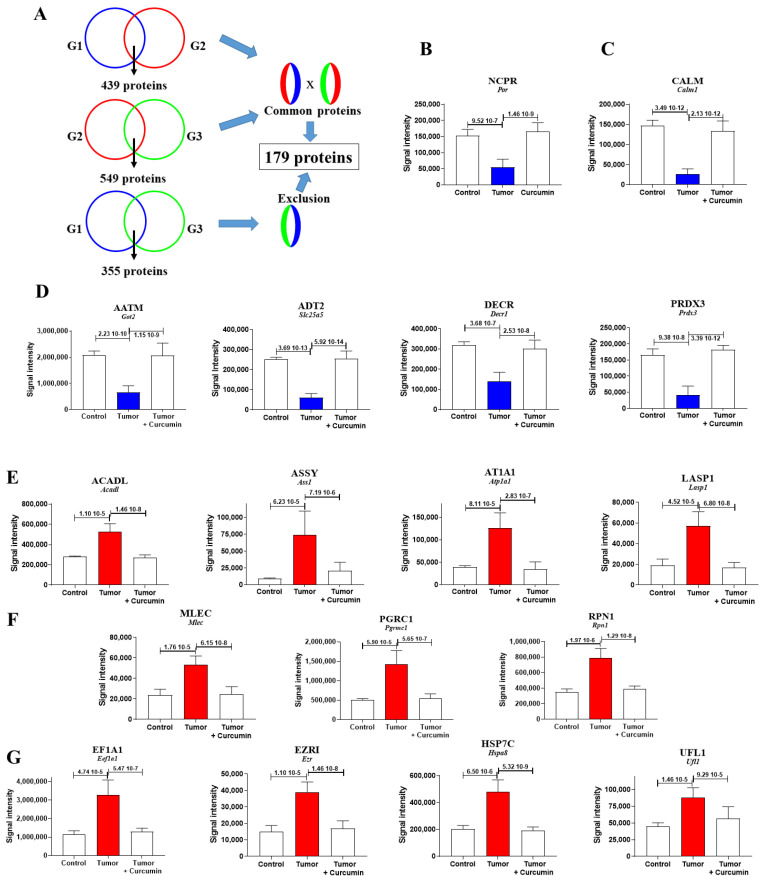
Liver biomarkers exhibiting the most dramatic quantitative changes under metastatic colonization by M5-T1 cells. (**A**) Diagram of the method used to identify the 179 biomarkers of interest. G1 = group of normal rats, G2 = group of untreated rats, G3 = group of curcumin-treated rats. (**B**–**D**) Biomarkers exhibiting the most dramatic changes within the 61 proteins presenting a decreased abundance, located in endoplasmic reticulum (**B**), cytoplasm and cytoskeleton (**C**), and mitochondria (**D**). (**E**–**G**) Biomarkers exhibiting the most dramatic changes within the 118 proteins presenting an increased abundance, located in mitochondria, cytoplasm or plasma membrane (**E**), endoplasmic reticulum (**F**), or present in multiple subcellular compartments (**G**).

**Figure 4 cancers-12-03384-f004:**
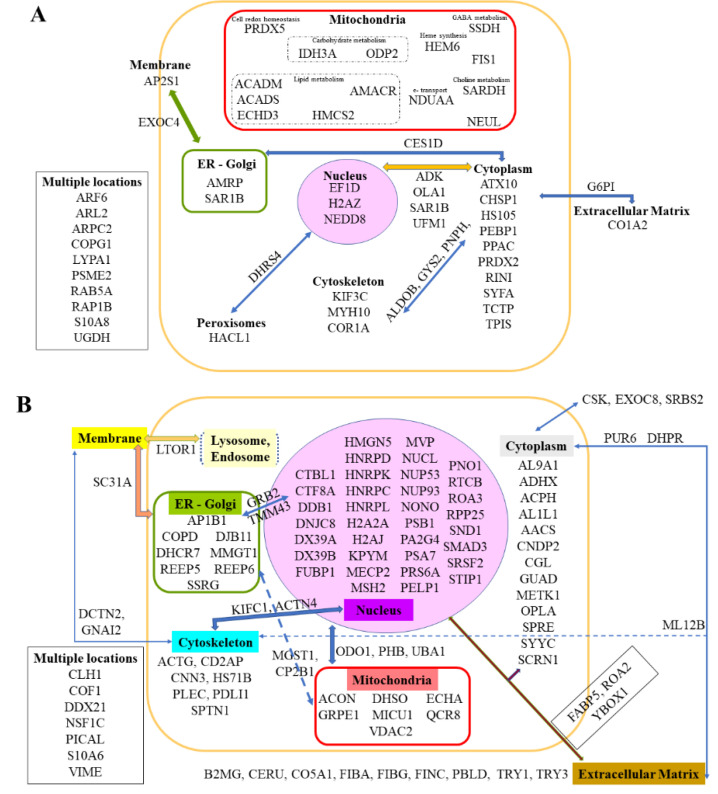
Additional biomarkers of interest showing abundance changes under metastatic colonization. (**A**) Liver biomarkers exhibiting significant decreased abundance in G2 vs. G1 and G3 vs. G2 but not in G3 vs. G1. (**B**) Liver biomarkers exhibiting significant increased abundance in G2 vs. G1 and G3 vs. G2 but not in G3 vs. G1. Each protein of interest is represented by its abbreviation and location (according to Uniprot.org). For proteins located in several subcellular compartments or extracellular matrixes, links between these are represented by arrows. The full names and equivalent human genes that encode the corresponding proteins are provided in Appendix A.

**Figure 5 cancers-12-03384-f005:**
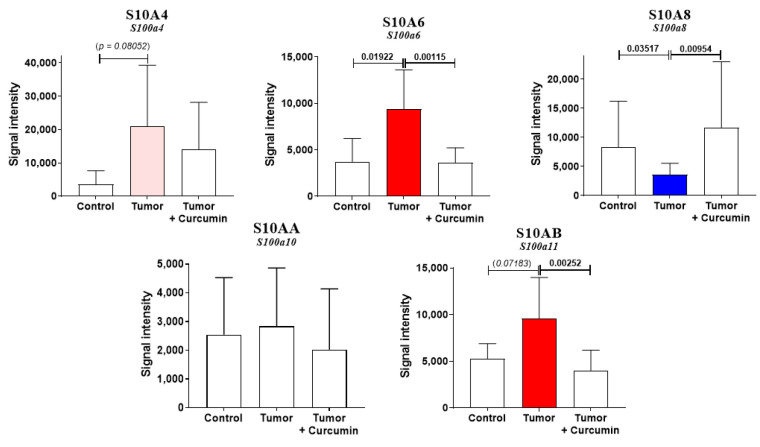
S100 differential abundance proteins. Comparison of protein abundances between the three groups of rats (significant increases in red and decreases in blue, with *p*-values indicated above the bars). Blank bars correspond to the absence of significant differences between groups. The *p*-values in italics and/or bars in light red correspond to tendencies (*p* < 0.09).

**Figure 6 cancers-12-03384-f006:**
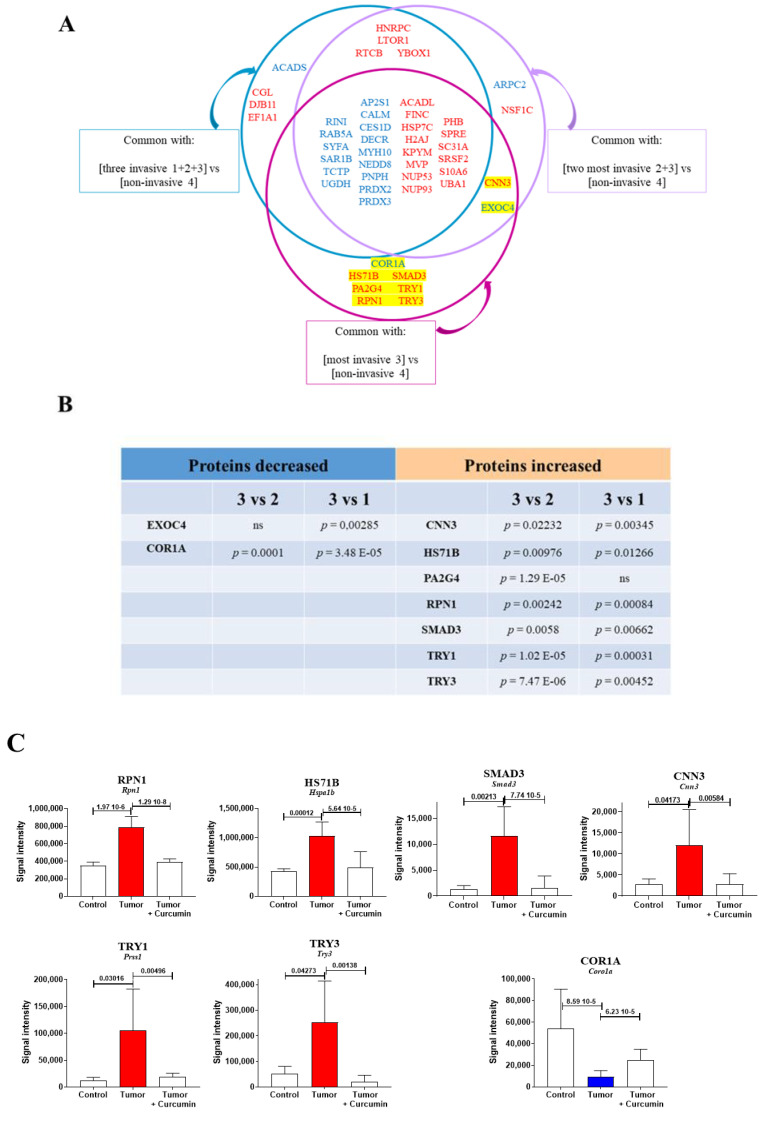
Common biomarkers of liver colonization by M5-T1 cells and the M5-T1 tumor. (**A**) Schematic representation of the method used to identify biomarkers common to the liver colonization by M5-T1 tumor cells and invasive rat mesothelioma tumors. The lists of proteins common to each condition are written in blue (for decrease) and red (for increase). The codes describing each invasive rat mesothelioma tumors were: 1 = F4-T2, 2 = F5-T1, and 3 = M5-T1 (4 = noninvasive M5-T2). As in Figure 3, Figure 4 and Figure 5, the equivalent human genes that encode the proteins represented by abbreviations (for the rat) are given in Appendix A. (**B**) The *p*-values for the nine biomarkers identified in (**A**), highlighted in yellow (MarkerView statistical analysis) when comparing abundance changes in 3 vs. 1 and 3 vs. 2 (same codes for the rat tumors as above). (**C**) Comparison of protein abundances between the three groups of rats (significant increases in red and decreases in blue, with *p*-values indicated above bars) for the seven biomarkers satisfying the condition *p* < 0.05 in the two tests described in (**B**).

**Figure 7 cancers-12-03384-f007:**
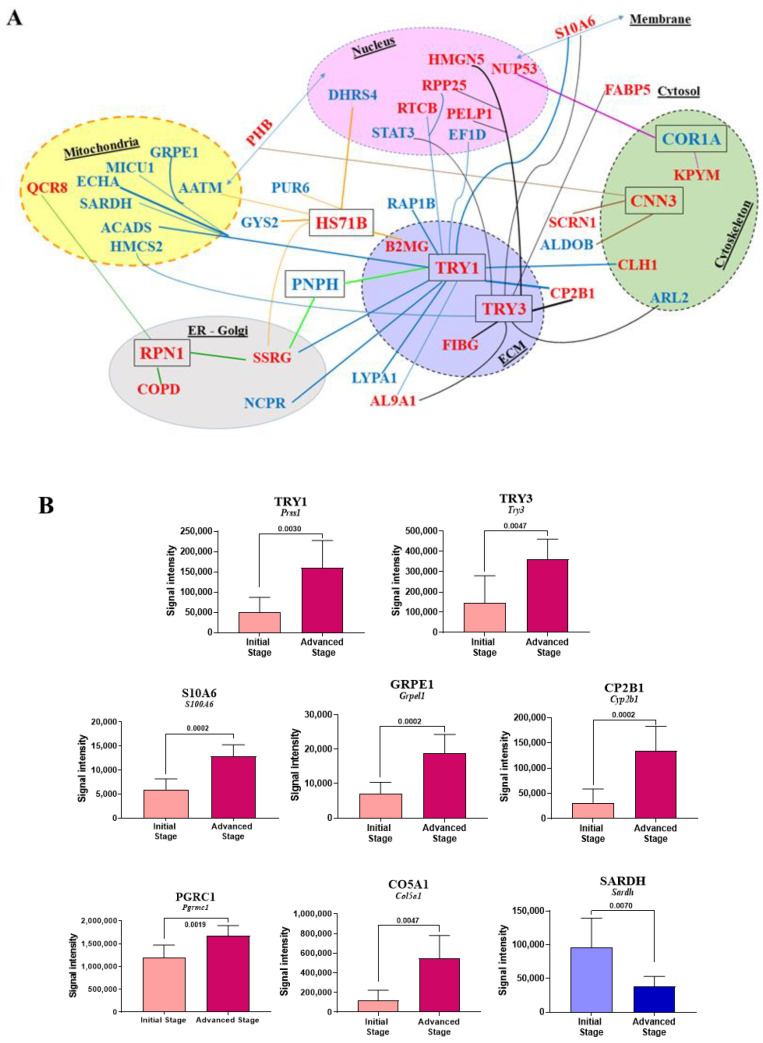
Main correlations observed between biomarkers identified in Section 2.2, Section 2.3, Section 2.4 and Section 2.5. (**A**) Subcellular locations of proteins of interest are represented by different colors (except for cytoplasm/cytosol). Proteins distributed between several subcellular compartments are labeled with double-headed arrows. The *p*-values < 0.01 observed between PNPH, the 179 biomarkers described in Section 2.3 (see Figure 3 and Figure 4), and each of the seven proteins finally identified in Figure 7C are symbolized by lines of different colors. (**B**) Most significant protein abundance changes in the liver between the two stages of M5-T1 tumor development defined in Section 4.2 and illustrated in Figure 1B,C.

**Table 1 cancers-12-03384-t001:** Statistical analysis of differential protein abundance between groups.

Protein	Initial St. (G2) /Control (G1)	Advanced St. (G2) /Control (G1)	+ Curcumin (G3) /Initial St. (G2)	+ Curcumin (G3) /Advanced St. (G2)
FETUA	*ns (0.3677)*	**0.0081**	**0.0104**	**0.0002**
IRGM	**0.0283**	**0.0040**	*ns (0.1605)*	*ns (0.2345)*
TRY1	0.0727	**0.0040**	0.0830	**0.0002**
TRY3	*ns (0.5697)*	**0.0040**	**0.0011**	**0.0002**
ACADS	**0.0727**	**0.0081**	*ns (0.1049)*	**0.0047**
CHSP1	**0.0040**	**0.0040**	**0.0002**	**0.0002**
SARDH	*ns (0.1535)*	**0.0485**	*ns (0.1304)*	**0.0104**
ADHX	**0.0485**	**0.0283**	0.0830	**0.0070**
AL1L1	0.0727	**0.0040**	*ns (0.2345)*	**0.0030**
CO5A1	*ns (0.1535)*	**0.0162**	**0.0281**	**0.0006**
CP2B1	*ns (0.5697)*	**0.0040**	*ns (0.1949)*	**0.0002**
DNJC8	*ns (0.2141)*	**0.0040**	*ns (0.3823)*	**0.0011**
DX39A	*ns (0.6828)*	*ns (0.1091)*	**0.0104**	*ns (0.8785)*
FIBG	**0.0162**	**0.0040**	**0.0003**	**0.0002**
GRPE1	*ns (0.1091)*	**0.0040**	*ns (0.2786)*	**0.0002**
MMGT1	*ns (0.2828)*	**0.0162**	*ns (0.1049)*	**0.0019**
PDLI1	**0.0485**	**0.0040**	**0.0148**	**0.0011**
PELP1	**0.0081**	**0.0040**	*ns (0.1049)*	**0.0019**
PGRC1	**0.0040**	**0.0040**	**0.0003**	**0.0002**
RTCB	**0.0040**	**0.0040**	**0.0019**	**0.0006**
S10A6	*ns (0.2141)*	**0.0081**	**0.0104**	**0.0002**

Proteins for which abundance decrease or increase within the G2 group (“Tumor”) is magnified with tumor progression. Nonsignificant differences (*ns*) are indicated in italics with calculated *p* values in smaller characters in brackets. Tendencies are indicated in normal type, and bold type was used to draw attention to *p* values < 0.05. Full names of proteins are given in Appendix A.

**Table 2 cancers-12-03384-t002:** Statistical analysis of differential protein abundance between groups.

Protein	Initial St. (G2) /Control (G1)	Advanced St. (G2) /Control (G1)	+ Curcumin (G3) /Initial St. (G2)	+ Curcumin (G3) /Advanced St. (G2)
AP2S1	**0.0040**	**0.0040**	**0.0003**	**0.0030**
ARL2	**0.0162**	*ns (0.1091)*	**0.0003**	**0.0148**
DJB11	**0.0040**	**0.0162**	**0.0003**	**0.0030**
ECHA	**0.0040**	**0.0081**	**0.0002**	**0.0011**
FABP5	**0.0040**	**0.0283**	**0.0002**	**0.0070**
FINC	**0.0040**	**0.0485**	**0.0019**	*ns (0.1605)*
H2AJ	**0.0040**	**0.0081**	**0.0030**	*ns (0.1949)*
HMCS2	**0.0040**	*ns (0.2828)*	**0.0002**	*ns (0.3823)*
HNRPK	**0.0040**	*ns (0.1091)*	**0.0002**	0.0830
IDH3A	**0.0081**	*ns (0.1535)*	**0.0006**	**0.0047**
LYPA1	**0.0040**	**0.0040**	**0.0002**	**0.0003**
NCPR	**0.0040**	**0.0040**	**0.0002**	**0.0002**
PA2G4	**0.0040**	**0.0162**	**0.0002**	0.0650
PICAL	**0.0081**	**0.0485**	**0.0011**	**0.0379**
PPAC	**0.0040**	**0.0040**	**0.0070**	**0.0104**
SRSF2	**0.0040**	*ns (0.1091)*	**0.0006**	*ns (0.2345)*
VIME	**0.0040**	**0.0162**	**0.0019**	*ns (0.5054)*

Proteins for which abundance decrease or increase within the G2 group (“Tumor”) does not follow tumor progression. Nonsignificant differences (*ns*) are indicated in italics with calculated *p* values in smaller characters in brackets. Tendencies are indicated in normal type, and bold type was used to draw attention to *p* values < 0.05. Full names of proteins are given in Appendix A.

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
