# Peer review of "Curcumin Treatment Identifies Therapeutic Targets within Biomarkers of Liver Colonization by Highly Invasive Mesothelioma Cells—Potential Links with Sarcomas"

_cancers, 2020, doi:10.3390/cancers12113384_

Round 1

Reviewer 1 Report

The study is of  high interest, and the results and discussion of findings is appropriately supported by literature. The findings are important. However, there are some problems.

authirs state that 'We previously demonstrated that rats given four injections of 1.5 mg/kg of curcumin on days 7, 76 9, 11, and 14 after tumor challenge presented a significant reduction in their mean total tumor mass 77 compared with untreated rats [17].' How was the curcumin injected? Ip, im, iv? Previous work  is quoted but it would be reasonable to at least state where the injections was.  The half-life of curmin in that context, and metabolism should be taken into account- the half life after iv injection is very short ( Reference Hocking, A, 2020) so how is the effect exerted? And the fact that much of the metabolism takes place in the liver would be of interest, too.

Chracterisation of infiltrating lymphocytes by IHC (C4, CD 8 at the very least) would be desirable. A major concern is the Figure 1D- the yellow arrow does NOT point at a tumour cell,  this is a bile duct...... in each picture. A picture clearly showing tumour would be very important. The location of the lymphocytes in the parenchyma would be ok, but we need to feel confident that the researchers have identified tumour correctly.

FIGURE 1 D MUST be changed in my opinion, maybe the authors accidentally used the wrong image.

Figure 7 is not readable. May be a supplementary figure could be used?

Author Response

The study is of high interest, and the results and discussion of findings is appropriately supported by literature. The findings are important. However, there are some problems.
authirs state that 'We previously demonstrated that rats given four injections of 1.5 mg/kg of curcumin on days 7, 76 9, 11, and 14 after tumor challenge presented a significant reduction in their mean total tumor mass 77 compared with untreated rats [17].' How was the curcumin injected? Ip, im, iv? Previous work is quoted but it would be reasonable to at least state where the injections was.
We have added in the new Fig. 1 (in A, line 109) a scheme representing the experimental design and details of the treatment procedure have also been inserted in paragraph 4.2. of the Materials and Methods section (lines 457-462).
The half-life of curmin in that context, and metabolism should be taken into account- the half life after iv injection is very short ( Reference Hocking, A, 2020) so how is the effect exerted? And the fact that much of the metabolism takes place in the liver would be of interest, too.
The paper published by Hocking et al. has been referenced in the last paragraph of the discussion, and a sentence referring to their work (in link with investigations on curcumin pharmacokinetics) is also included (lines 427-430).
Chracterisation of infiltrating lymphocytes by IHC (C4, CD 8 at the very least) would be desirable.
Thank you for suggesting us to add these illustrations. Although some were already provided in the supplementary Fig. S5 of our paper published in Oncotarget 2017 (reference [17] of the present manuscript), we include here new unpublished IHC photographs for that purpose in Figs 1J-K (line 111).
A major concern is the Figure 1D- the yellow arrow does NOT point at a tumour cell, this is a bile duct...... in each picture. A picture clearly showing tumour would be very important.
We are sorry about this confusion which came from the fact that M5-T1 tumor cells frequently invaded the vicinity of bile ducts. So, to deal with that confusion, please find in the new version of Fig. 1 (in H) three photographs representative of different rats of the “Tumor” (G2) group taken close to the tumor front where invasive M5-T1 tumor cells are clearly visible (between lines 110 and 111).
The location of the lymphocytes in the parenchyma would be ok, but we need to feel confident that the researchers have identified tumour correctly.
FIGURE 1 D MUST be changed in my opinion, maybe the authors accidentally used the wrong image.
Figure 7 is not readable. May be a supplementary figure could be used?
Thank you for this suggestion, this Figure (previous Fig. 7B) has been deleted in the corrected version of the manuscript (line 283), and is now included in the supplementary materials (Fig. S2).

Reviewer 2 Report

  • This  is a  relevant study and a well-written manuscript  entitled  “Curcumin treatment identifies therapeutic targets within biomarkers of liver colonization by highly invasive mesothelioma cells — potential links with sarcomas”  By  Daniel L. Pouliquen *, Alice Boissard, Cécile Henry, StéphanieBlandin, Pascal Richomme, Olivier Coqueret and Catherine Guette.
    • The authors started by analyzing 3 different  groups of  rats in which non transformed liver cells derived from rats  affected by sarcomatoid mesothelioma  (M5-T1) , an aggressive tumour characterized previously as having  dysregulated  MHC class I presentation , were either untreated  (no Curcumin (G2) or curcumin-treated in rats without hepatic metastases (G3) as compared to syngeneic normal  rat liver cells (G1). They used proteomic quantitative analyses to further demonstrate that liver microenvironment might determine the metastatic potential of the tumour and identified  by sequential substraction analyses few key players. The data are consistent and presented in clear-cut conceived figures
  • The comparative Swath-MS convincely identify 12 biomarkers
  • The study shows in Curcumin-treated rats an increased number in circulating CD8+ T cells, characterized the localization, morphology,  and size  of the lymphocytes present in liver parenchyma, as compared to the untreated  tumour-bearing G2 rats and G1  controls. Further comparisons  to tumours /M5T1 cells  and nonmetastatic liver tumours showed 7 biomarkers that correlate with immune modulation, antigen presentation  or cytoskeleton (calponin 3) increase in controls or corionin 1A  decrease in untreated  rats.
  • This reviewer  essentially  has  ranked this MS as an excellent study and thereby has only minor points to raise
  • Please could you better clarify  in the “Abstract “  the tumour model
  • (M5-T1)   characterisitcs’? Possibly describing  briefly your previous  main findings concerning  M5-T1 invasiveness .
  • In the Abstract  sentence Line 25 « as » could be changed in « being»?
  • This reviewer greatly appreciated the efforts in describing the enzimatic activity / functions of the most relevant players you have identifiied  here.
  • Tumorigenesis is dependent upon tumor microenvironment to favor tumour progression and dissemination and  it is of outmost importance to identifiy prognostic markers and novel therapeutical options.
  • Purine salvage pathway and  PNP deficiency in humans leads to an impairment of T-cell function,  please citations should be inserted  concerning those related to immunodeficiency and /or  a general review should be cited such as  « Purine nucleoside phosphorylases: properties, functions, and clinical aspects » by A Bzowska 1, E Kulikowska, D Shugar
  • Similarly, S100 proteins are a class of calcium binding  pro inflammatory Proteins well described in the review by Gilson, Skaar and Chazin which should be  quoted.
  • Ceruplasmin is  a Ferroxidase enzyme which is the major copper-carrying protein in the blood with its  copper-dependent oxidase activity being associated with iron oxidation to support iron transport , besides being a stress acute phase marker.

  • In the Material and Methods section please quote in the title the « experimental  procedures for the in vivo manipulations « in previous  histological section.
  • Finally, please  would be worth citing the review « Curcumin as anticancer agent in malignant  mesothelioma »   by Baldi and co-authors published in International J.  Molecular Sciences which is  relevant being focussed in mesothelioma treatments.
  • To conclude, any perspectives for clinical translation of your key observations are welcome.

Author Response

This is a relevant study and a well-written manuscript entitled “Curcumin treatment identifies therapeutic targets within biomarkers of liver colonization by highly invasive mesothelioma cells — potential links with sarcomas” By Daniel L. Pouliquen *, Alice Boissard, Cécile Henry, StéphanieBlandin, Pascal Richomme, Olivier Coqueret and Catherine Guette.
o The authors started by analyzing 3 different groups of rats in which non transformed liver cells derived from rats affected by sarcomatoid mesothelioma (M5-T1) , an aggressive tumour characterized previously as having dysregulated MHC class I presentation , were either untreated (no Curcumin (G2) or curcumin-treated in rats without hepatic metastases (G3) as compared to syngeneic normal rat liver cells (G1). They used proteomic quantitative analyses to further demonstrate that liver microenvironment might determine the metastatic potential of the tumour and identified by sequential substraction analyses few key players. The data are consistent and presented in clear-cut conceived figures
• The comparative Swath-MS convincely identify 12 biomarkers
• The study shows in Curcumin-treated rats an increased number in circulating CD8+ T cells, characterized the localization, morphology, and size of the lymphocytes present in liver parenchyma, as compared to the untreated tumour-bearing G2 rats and G1 controls. Further comparisons to tumours /M5T1 cells and nonmetastatic liver tumours showed 7 biomarkers that correlate with immune modulation, antigen presentation or cytoskeleton (calponin 3) increase in controls or corionin 1A decrease in untreated rats.
• This reviewer essentially has ranked this MS as an excellent study and thereby has only minor points to raise
• Please could you better clarify in the “Abstract “ the tumour model
• (M5-T1) characterisitcs’? Possibly describing briefly your previous main findings concerning M5-T1 invasiveness .
In the abstract, we just change one word, as we could not exceed 200 words… and we added two sentences for that purpose in the last paragraph of the introduction (lines 69-75).
• In the Abstract sentence Line 25 « as » could be changed in « being»?
This word was changed (now line 31).
• This reviewer greatly appreciated the efforts in describing the enzimatic activity / functions of the most relevant players you have identifiied here.
• Tumorigenesis is dependent upon tumor microenvironment to favor tumour progression and dissemination and it is of outmost importance to identifiy prognostic markers and novel therapeutical options.
• Purine salvage pathway and PNP deficiency in humans leads to an impairment of T-cell function, please citations should be inserted concerning those related to immunodeficiency and /or a general review should be cited such as « Purine nucleoside phosphorylases: properties, functions, and clinical aspects » by A Bzowska 1, E Kulikowska, D Shugar
Thank you very much for providing us the opportunity to cite this important review (see the second paragraph of the discussion lines 310-312).
• Similarly, S100 proteins are a class of calcium binding pro inflammatory Proteins well described in the review by Gilson, Skaar and Chazin which should be quoted.
We have added this reference in the fifth paragraph of the discussion, lines 348-349.
• Ceruplasmin is a Ferroxidase enzyme which is the major copper-carrying protein in the blood with its copper-dependent oxidase activity being associated with iron oxidation to support iron transport , besides being a stress acute phase marker.
We have added part of a sentence lines 146-147 in order to give this additional information.
• In the Material and Methods section please quote in the title the « experimental procedures for the in vivo manipulations « in previous histological section.
We have added this title as a subheading of the paragraph 4.3. in the Materials and Method section (line 464).
• Finally, please would be worth citing the review « Curcumin as anticancer agent in malignant mesothelioma » by Baldi and co-authors published in International J. Molecular Sciences which is relevant being focussed in mesothelioma treatments.
We agree with your suggestion, this important reference in the field is included in the first sentence of the last paragraph of the discussion (lines 424-427).
• To conclude, any perspectives for clinical translation of your key observations are welcome.
Thank you, this prospect is just mentioned also in the last paragraph of the discussion (line 427).

Reviewer 3 Report

The paper by Poliquinar et al., showed some interesting results obtained by an innovative approach, nevertheless additional data are needed to clearly understand the potentiality of their findings.

First of all, in the results session more a detailed description of the experimental design had to be given (not only the references to the pervious papers). Curcumin doses, schedule of administration, timing of sample collection, time from the last curcumin dose and n° of rats without livers metastasis over n° of treated rats etc... are mandatory information for the correct interpretation of the results.

Second, to asses if the effect of curcumin is directed on liver tissue or mediated by cancer cell, proteomic analysis should be performed in healthy rats treated with curcumin. Similarly, to understand if the change in the protein expression is due to the treatment or to the presence of the metastasis in the liver parenchyma and if it is responsible for tumor dissemination, the same determination has to be done in liver tissues form curcumin treated mice that have developed metastasis.

Line 374 states: “nine rats bearing M5-T1 tumors at different stages of development”. What does it mean? Different times from tumor inoculum? How these different stages were considered in the proteomic analysis? If a sufficient number of liver has been collected at different stages of colonization the kinetic of protein expression would be really interesting to define their potential role as therapeutic target.

Author stated in the introduction that they have previously identified S10A4 (line 62) as biomarker of all stages of MM development but according to fig 5 the increment of this protein in the liver of tumor bearing rats is not significant despite the parenchyma was colonized by MM cells. Please comment this result.

Paragraph 2.5: common biomarkers of liver colonization by M5-T1 cells and the M5-T1 tumor.  It’s not clear the correspondence between the title and the data reported in the paragraph were a comparison between the markers of liver colonization of different rat mesothelioma tumors is done. Moreover, no information are provided about the source of the samples.

With the exception of figure 1 all the others do not report the error bars. Please add.

In general, a better description of the experimental plans, aims and results is required to evaluate the relevance of the reported findings.

Author Response

The paper by Poliquinar et al., showed some interesting results obtained by an innovative approach, nevertheless additional data are needed to clearly understand the potentiality of their findings.
First of all, in the results session more a detailed description of the experimental design had to be given (not only the references to the pervious papers). Curcumin doses, schedule of administration, timing of sample collection, time from the last curcumin dose and n° of rats without livers metastasis over n° of treated rats etc... are mandatory information for the correct interpretation of the results.
We are sorry for these missing information, and we have included in Figure 1 (A) of the corrected manuscript a scheme explaining the experimental procedure used for collecting and selecting samples of the two groups “Tumor” (G2) and “Tumor + Curcumin” (G3), between lines 109 and 110. Some lines of explanation have also been inserted at the beginning of the Results section (lines 88-93), and in the paragraph 4.2 of the Materials and Method section (lines 451-462).

Second, to asses if the effect of curcumin is directed on liver tissue or mediated by cancer cell, proteomic analysis should be performed in healthy rats treated with curcumin. Similarly, to understand if the change in the protein expression is due to the treatment or to the presence of the metastasis in the liver parenchyma and if it is responsible for tumor dissemination, the same determination has to be done in liver tissues form curcumin treated mice that have developed metastasis.
Unfortunately, we cannot provide this additional information as we did not include this group in the study, and it is impossible to do it presently as I moved to another team and stopped working on laboratory rodents about 2 years ago (tissue samples for this study were collected several years ago). Moreover, may I mention that I don’t think the study of curcumin in healthy rats will help significantly as several lines of evidences (from preliminary investigations referred as Experiment 2 in Oncotarget 8(34), 57552-73(2017), Figs 7C and 7E) suggest the pharmacokinetics and metabolization in the liver of curcumin injected i.p. would be different (probably longer and reduced, respectively) in tumor-bearing rats compared with healthy rats. One observation related to this point was the yellowish color of tumors and adipose tissue collected from rats necropsied at least one day after the last injection of curcumin i.p. (D21 and D26 after tumor challenge), probably related to bile ducts obstruction by metastatic tumor cells. It is also probable that pharmacokinetics and metabolic data will depend on the important level of variation in total tumor mass and extent of metastases within a group of rats even when they are necropsied the same day after tumor inoculum (individual differences).

Line 374 states: “nine rats bearing M5-T1 tumors at different stages of development”. What does it mean? Different times from tumor inoculum? How these different stages were considered in the proteomic analysis? If a sufficient number of liver has been collected at different stages of colonization the kinetic of protein expression would be really interesting to define their potential role as therapeutic target.
Yes, “tumors at different stages of development” refers to different times from tumor inoculum (done at Day 0 = D0 in the scheme in Fig. 1A), D16 to D21, but also to individual differences in tumor growth rate observed between rats of the same group and receiving the same amount of M5-T1 tumor cells. In order to illustrate two opposite steps in this tumor development, histological features of the liver, tumor front and invasion of organs are provided in Figs 1B-C, and the same is also included for samples belonging to our M5-T1 tumor bank but excluded from the study (Fig. 1D), due to extended invasion of the liver and infiltration of tumor cells in the parenchyma that did not allow to accurately separate the liver tissue from the tumor for proteomic analysis.

Author stated in the introduction that they have previously identified S10A4 (line 62) as biomarker of all stages of MM development but according to fig 5 the increment of this protein in the liver of tumor bearing rats is not significant despite the parenchyma was colonized by MM cells. Please comment this result.
In our previous study on S10A4 (reference [16]), we observed an important increase in the abundance of S10A4 in the spleen (instead of the liver in the present manuscript). Herein, we observed a 6 fold increase (in average) in the abundance of this protein in the liver of untreated tumor rat compared with normal rats. The fact that this difference corresponded to a small tendency (with p = 0.08), is explained by an important dispersion of the values within this group of tumor-bearing rats. In addition, in our previous study on spleens, the differences were analyzed on more rats (10 in each group).

Paragraph 2.5: common biomarkers of liver colonization by M5-T1 cells and the M5-T1 tumor. It’s not clear the correspondence between the title and the data reported in the paragraph were a comparison between the markers of liver colonization of different rat mesothelioma tumors is done. Moreover, no information are provided about the source of the samples.
We are sorry for this lack of precision. We have added a sentence (line 238) to explain the source of these lists. They refer to one of our previous work [reference [15]) aiming at investigating invasiveness differences between the four types of tumors (not liver colonization by the different tumors because the non-invasive M5-T2 tumor cells never infiltrates the liver, and the F4-T2 tumor cells only partly).

With the exception of figure 1 all the others do not report the error bars. Please add.
Error bars have been added in all figures.

In general, a better description of the experimental plans, aims and results is required to evaluate the relevance of the reported findings.
We hope the different figures and text added in the revised version of our manuscript will help better understand our findings.

Reviewer 4 Report

Very interisting and well adressed work on a topic not much studied (curcumin and mesothelioma). I would suggest to add  a few lines on the role of curcumin in malignant mesothelioma (Baldi A, De Luca A, Maiorano P, D'Angelo C, Giordano A. Curcumin as an Anticancer Agent in Malignant Mesothelioma: A Review. Int J Mol Sci. 2020 Mar 7;21(5):1839. doi: 10.3390/ijms21051839.)

Author Response

Thank you for this suggestion, we have mentioned this important reference in the first sentence of the last paragraph of the discussion (lines 425-427).

Round 2

Reviewer 3 Report

The revised version of the paper by Pouliquen at al.  answers the revisor requests. The main issue remains the experimental scheme that is not fully adequate to draw reliable conclusions. According to the additional information reported in fig 1, the G2 group (tumor) is composed of 8 rats sacrificed at different times from tumor inoculum and exhibiting different stages of tumor development, resulting in a very heterogeneous group but this condition was not taken into account in the data analysis. How many rats for each stage were analyzed? Do the levels of the selected proteins change between the initial and advanced stages? This information is critical to understand if the results obtained in group 3 (curcumin-treated rats) are due to the effect of curcumin or to the selection of the tumor-free liver. Would the effect of curcumin have been present also in the two rats that experience liver metastasis despite treatment? Are samples from these rats still available?

Author Response

The revised version of the paper by Pouliquen at al. answers the revisor requests. The main issue remains the experimental scheme that is not fully adequate to draw reliable conclusions. According to the additional information reported in fig 1, the G2 group (tumor) is composed of 8 rats sacrificed at different times from tumor inoculum and exhibiting different stages of tumor development, resulting in a very heterogeneous group but this condition was not taken into account in the data analysis. How many rats for each stage were analyzed?

We thank the reviewer for these crucial questions and for giving us the opportunity to provide an improved and more complete version of our manuscript.
There were four rats representing the initial stage and four others corresponding to the advanced stage of the G2 group (“tumor”). This information has been included in the text in section 4.2. of the Materials & Methods (lines 478-480), and section 2.6. of Results (2nd paragraph, line 271).
Do the levels of the selected proteins change between the initial and advanced stages? This information is critical to understand if the results obtained in group 3 (curcumin-treated rats) are due to the effect of curcumin or to the selection of the tumor-free liver.

To assess that point, we have completed the second paragraph of section 2.6. of the Results (lines 268 to 282) and added a third one (lines 283-291). For that purpose, changes in the levels of all proteins described in sections 2.2 to 2.4. of Results (and Figs 2 to 5) between the two stages have been analyzed, the results being presented in supplementary Table S3. The most important changes were also included in a new corrected version of Fig. 7B, line 292.
As explained in the text line 283 “as abundance variations within the G2 group (“Tumor”) could influence the previously described effect of curcumin”, with the aim to complete our response to the important question raised by the reviewer above, we included in the two tables 1 and 2 (lines 303-308 and 309-314), the results of the statistical analyses of four comparisons, giving p values for 21 proteins showing an enhancement of protein abundance change with the tumor development stage, or the inverse situation (17 proteins). We also completed the sentences previously added in the revised version of the discussion (lines 399 to 406), including a new reference.
Would the effect of curcumin have been present also in the two rats that experience liver metastasis despite treatment? Are samples from these rats still available?

Unfortunately, we had not enough time to check if some liver tissue would be available from the paraffin-embedded tissue samples of these two rats. Due to the Covid pandemics and a new lockdown in our country, it will take months to see if some proteomic data could be obtained from these samples to try to partly respond to this interesting question. Moreover, as shown in our article published in Oncotarget (2017) 8(34), 57552-57573, in Fig. 9A p. 57563, there was an important dispersion in the remaining total tumor mass for these two rats (as for the group of rats with untreated tumors).
So, to sum up, I think that to fully assess this question, new experiments that could hopefully be designed in the future might include an important number of animals for that purpose (at least 18), as these two rats represented only one third of the size of the curcumin-treated rats group.

Round 3

Reviewer 3 Report

the mechanism of curcumin on liver parenchyma is not fully supported by the data but given the impossibility to obtain more samples on curcumin-treated samples it can not further investigated. I would suggest reducing the emphasis given to this in the first lines of the conclusion